# Shifting Gears in Precision Oncology—Challenges and Opportunities of Integrative Data Analysis

**DOI:** 10.3390/biom11091310

**Published:** 2021-09-04

**Authors:** Ka-Won Noh, Reinhard Buettner, Sebastian Klein

**Affiliations:** 1Institute for Pathology, Faculty of Medicine and University Hospital Cologne, University of Cologne, 50937 Cologne, Germany; ka-won.noh@uk-koeln.de (K.-W.N.); Reinhard.Buettner@uk-koeln.de (R.B.); 2Gerhard-Domagk-Institute of Pathology, University Hospital Münster, 48149 Münster, Germany

**Keywords:** precision oncology, integrative data analysis, cancer genomics, digital pathology, biomarker, image analysis, immunotherapy

## Abstract

For decades, research relating to modification of host immunity towards antitumor response activation has been ongoing, with the breakthrough discovery of immune-checkpoint blockers. Several biomarkers with potential predictive value have been reported in recent studies for these novel therapies. However, with the plethora of therapeutic options existing for a given cancer entity, modern oncology is now being confronted with multifactorial interpretation to devise “the best therapy” for the individual patient. Into the bargain come the multiverse guidelines for established and emerging diagnostic biomarkers, as well as the complex interplay between cancer cells and tumor microenvironment, provoking immense challenges in the therapy decision-making process. Through this review, we present various molecular diagnostic modalities and techniques, such as genomics, immunohistochemistry and quantitative image analysis, which have the potential of becoming powerful tools in the development of an optimal treatment regime when analogized with patient characteristics. We will summarize the underlying complexities of these methods and shed light upon the necessary considerations and requirements for data integration. It is our hope to provide compelling evidence to emphasize on the need for inclusion of integrative data analysis in modern cancer therapy, and thereupon paving a path towards precision medicine and better patient outcomes.

## 1. Introduction

Within the medical field, more specifically in oncology, treatments have shifted from a “one-size-fits-all” therapy to individualized treatment strategies for a given patient, supporting the notion of intratumoral and intertumoral heterogeneity. Consequently, this requires state-of-the-art diagnostic approaches along with centralized treatment efforts using digitized, scalable, and standardized information processing.

Historically, with the discovery of oncogenes, dedicated scientists have identified molecules that drive cancer initiation and progression. However, nowadays, its value has expanded to hold predictiveness in stratification of patients for selection of treatments [1]. For instance, overexpression of HER2 on tumor cell surfaces has been proposed as a potent therapeutic target [2,3] and its inhibition has led to clinical responses [4].

To mine for such biomarkers, specific diagnostic assays are conducted based on the cancer type and patient characteristics. Diagnostic biomarkers are regularly coupled with therapeutic drugs; for instance targeting specific oncogenic alterations is a known concept as companion diagnostics [5,6]. Correctly identifying the molecular characteristics of the given cancer at a certain level of confidence is key in the process. Although it may be fortunate in terms of therapeutic advances, from a logistics perspective, there has been a growing number of concerns regarding the vast number of targets that have been identified throughout several different cancer entities. Thus, scant tissue samples may become confronted with large number of assays. Interestingly, decision support systems are already available that may guide therapy decision making for individual cancer patients, although these technologies still need further improvement [7]. However, given that resistance mechanisms of therapy that may occur, depending on the time of treatment and other patient-, entity-, and therapy-specific traits, specimens may have to undergo additional analyses [8,9]. For example, a prolonged course of therapy has been reported to give rise to persistent clonal variants by adding specific selective pressure [10]. Consequently, these resistance mechanisms may require additional sample processing throughout treatment, such as performing liquid biopsy analyses [11].

In parallel, there is an additional layer of complexity with the recent advances in immunotherapy, a novel treatment strategy allowing us to reprogram the tumor microenvironment (TME) towards pro-immunogenicity [12,13]. Here, therapeutics are administered that block natural immune-defense mechanisms to allow the immune system to alleviate anti-cancer activities. On one hand, there seems to be a lack of biomarkers that show both sensitivity and specificity to identify patients that would respond to immunotherapy. On the other hand, combinatorial treatment regimens with other immune-checkpoint blockers or different small molecule inhibitors in combination with other therapies are emerging [14].

With the following review, we will first introduce selected current and future diagnostic modalities being used in precision oncology and illustrate the elements that are needed to translate these diagnostics to a more integrative approach and factors that should be considered for a successful integration.

## 2. Current and Future Diagnostic Modalities in Precision Oncology

### 2.1. Sources of Genomics Data

Genomics, the study of genes and their functions, encompasses various aspects of the genome, such as genetics, transcriptomics, epigenetics, proteomics and more recently, dissection of the aforementioned aspects in single-cell resolution. By incorporating cancer genomics with clinical data, translational research is achieving higher levels of relevance and precision in the clinical care they provide to cancer patients [15]. A significant contribution to integrative analysis in oncology has been made by The Cancer Genome Atlas (TCGA) funded by the National Cancer Institute (NCI) [16,17].

For this review, a focused publication retrieval was done on the PubMed database by using keywords, such as precision oncology AND genomics OR molecular OR integrative OR multiomics.

#### 2.1.1. DNA-Sequencing

For decades, genetic alterations have been used to diagnose a variety of hereditary diseases that were previously undiagnosed [18,19,20]. However, in the era of modern oncology, genetic alterations are now being referred to guide medical care decisions [21]. Depending on the type of DNA input used in the sequencing process, DNA sequencing can reveal somatic and germline genetic mutations, insertions/deletions, amplifications, deletions, chromosomal copy number variations, gene fusions and other structural alterations [22,23,24].

There are several different genomic approaches that may be used for DNA-sequencing (Table 1). Whole-genome sequencing (WGS) allows researchers to examine both the exons (coding) and introns (non-coding) of the DNA, which roughly accounts to 3 billion base-pairs [25,26]. On the other hand, whole-exome sequencing (WES) captures and analyzes only the exonic regions (approximately 30 million base-pairs) of the genome [25,26]. This approach has the advantage of decreasing the cost and time compared to WGS, since exons are translated into functional proteins and their alterations are most likely to produce phenotypic changes. Targeted gene panels, which captures key genes or areas of interest, is another attractive option to a significant reduction in time and cost compared to WGS and WES. As a result, several targeted gene panels have already been adopted and used regularly in clinical settings [27,28,29]. Diagnostically, WGS, WES as well as targeted gene panels are also being used for detection of copy number variations, tumor mutational burden, homologous repair deficiency and genomic scarring patterns, as well as mutational signatures. Interestingly, all of these parameters may have an impact on selection of effective tumor therapies.

One of the most widely used area is pharmacogenomics, which analyzes how an individual’s genome will affect the therapeutic response [30]. For example, these alterations may entail identifying individual targetable variants or estimating tumor mutational load to predict response to treatment, such as immune-checkpoint inhibitors [31,32]. Another emerging field in genetics is the use of non-invasive circulating tumor DNA to monitor the treatment responses of the patients as well as to characterize the resistance mechanisms [33,34]. With the advancement of technology, the relevance of genetics in the clinical setting is predicted to increase more soon.

#### 2.1.2. RNA-Sequencing

The transcriptome is comprised of all RNA molecules in a single cell, or a group of cells, and it can be applied to all RNAs or only messenger RNA (mRNA). Sequencing of tumor RNA as well as its surrounding TME, such as stromal cells and the extracellular matrix, allows for a comprehensive characterization of transcriptome, such as gene fusions, gene signatures, small RNAs, spliced variants and allele-specific expression patterns [35,36]. Given the fact that clinical applications utilizing RNA expression in tumors are already being implemented in clinical practice, these applications have transformed the way patients are diagnosed and treated [37].

Like DNA-sequencing, genome-wide transcriptome can be assessed using RNA-sequencing (RNA-seq) as well as the array-based transcriptome by using targeted panels (Table 1). RNA-seq can generate a read depth of 10–30 million reads per sample on a high-throughput device, and this provides access to most of the transcriptome [38]. In work from our group, Klein et al. showed through transcriptomic profiling that advanced pleomorphic dermal sarcoma (PDS) cases generally illustrate inflamed TME [39,40], which has been reported to be associated with response to immune checkpoint inhibitors. As metastasized PDS cases are presented with limited therapeutic options, such as surgery, the fact that uncommon tumor entities may receive benefits from such prognostic applications is one exemplary case that shows the power of transcriptomic approach. Through this novel discovery, our team has reported, for the first time, two PDS cases who were successfully treated with pembrolizumab, an anti-PD-1 inhibitor [40].

Compared to RNA-seq, nCounter methods provide certain advantages, such as acceptance of poor-quality RNA, including formalin-fixed paraffin-embedded (FFPE) material, and that it is amplification free [41] (Table 1). Such advantages are what allowed its successful integration for clinical use, such as Prosigna, which predicts the recurrence risk of breast cancer patients [42].

#### 2.1.3. Epigenetics

Epigenetics entails studying changes in higher-order chromatin structure, as well as chemical modifications of DNA and/or histones, and epigenome mapping can be done with increasing precision nowadays. Epigenetic mechanisms can be divided into DNA methylation, histone modification and noncoding RNAs (ncRNAs) [43]. Since cancer is nowadays regarded to be both a genetic and an epigenetic disease, epigenetics is predicted to have a higher impact in the future of health care. Biomarkers, which are prognostic and/or predictive of therapeutic response, are currently a major area of interest in the clinical application of epigenetics. O^6^-methylguanine-DNA methyltransferase (MGMT) promoter methylation, a marker that predicts response to temozolomide chemotherapy in glioblastoma, is one example that is already implemented in a clinical setting [44,45].

Due to the advancement in technology, the discovery of epigenetic signatures that have the ability to distinguish neoplastic from matching normal cells is now possible with the inclusion of appropriate controls [46]. By taking into account the physiological epigenome change caused by aging, environmental stimuli or pathological disease, cell-type heterogeneity and the cell-of-origin can be investigated [47]. One recent example would include the study of epigenetic changes in mesenchymal stem cells during differentiation of cells-of-origin of solid tumors [48]. A deeper grasp of tumorigenesis and cancer progression can be therefore understood through epigenetic studies.

Other emerging areas in epigenetics compose of studying drug response related epigenetic alterations and dysregulation of microRNAs (miRNAs) leading to drug resistance [49,50,51]. Integration of large-scale studies of disease-associated epigenetic alterations (epigenome-wide associated studies; EWAS) and genome-wide associated studies (GWAS) data will be crucial for enhancing our knowledge in functional analysis [52].

#### 2.1.4. Proteomics

Proteomics enable the collecting of valuable information for treatment, diagnosis and pathophysiological mechanisms of various diseases that cannot be effectively characterized by epi/genetic- and transcriptomic-sequencing [53,54]. At a functional level, cancer is inherently a proteomic disease, and changes occur at the protein level through post-translational modifications and cellular signaling alterations [54,55]. Although broad proteomic analysis is still an emerging field in precision oncology, many of these signaling pathways are already established as U.S. Food and Drug Administration (FDA)-approved biomarkers [25,56,57]. For instance, prostate specific antigen (PSA) analysis in serum is now a standard of prostate cancer diagnosis, where it is accompanied with digital examination to assess whether prostate biopsies are necessary [58]. Some emerging clinical applications in proteomics include disease monitoring and improved patient stratification [55]. A detailed description on this topic on the future of proteomics in combination with genomics has been reviewed elsewhere [16].

Efforts to lower transplant rejection rates is one example. Despite the improvement in lowering transplant rejection rates through genetic profiling of HLA loci, up to 10% experience rejection or graft versus host disease (GVHD) [59,60]. It is predicted that personalized proteomic analysis of the peptidome may help in reducing this rate even further [61]. As a result, integration of proteomic analysis may help refine or modify epi/genetics and transcriptomics-guided personalized therapy.

#### 2.1.5. Single-Cell

Intratumor heterogeneity (ITH) is commonly displayed across various tumor entities and it is thought to be the key to understanding the complex process of metastasis and therapeutic resistance mechanisms [62]. Conventional sequencing technologies are conducted in bulk, where the effort to understand rare cell populations within the tumor poses a challenge. To overcome this aspect, high-throughput technologies profiling at a single-cell resolution—especially at the transcriptomic level—have been developed that offer the chance to explore phenotypically distinct subpopulations and states that can influence clinical outcomes, aid treatment strategies, or suggest novel therapeutic options [63,64].

Although considered as one of the latest additions to the family of genomics, it has already produced groundbreaking discoveries in immunology, neurobiology, and cancer biology [65,66,67]. One of the most profoundly influenced field is the study of TME. For example, a subset of quiescent cancer-associated fibroblasts (CAFs) was found to be enriched in the tissue culture of intrahepatic cholangiocarcinoma (ICC) after the blockade of placental growth factor (PIGF) [68]. Such a discovery may have been possible due to the robustness and granularity that this technique delivers. All these studies will prove to be an invaluable reference for future mechanistic research and the advancement of new immunotherapeutic or combinatorial strategies.

### 2.2. Image Analysis and Immunohistochemistry

#### 2.2.1. Quantitative Image Analysis Using Immunohistochemistry

Assessing biomarkers of (tumor) specimens on protein level using immunohistochemistry (IHC) has a long history, especially for routine diagnostics [69]. Advances in tissue processing using standardized automated staining devices allowed results of increasing accuracy and reproducibility [70]. Automated quantification techniques have been shown to provide increasing accuracy while decreasing inter- and intra-observer variabilities [71,72,73].

In parallel, the complexity to assess biomarkers of immune response has increased. PD-L1, the ligand of PD-1 that is one of the few biomarkers that may reveal response to immunotherapy in subpopulations, is assessed following strict criteria: membranous staining, staining of all intensities, staining of immune cell populations in certain proximity to cancer cells [74,75,76]. Here, assessing IHC biomarkers using sophisticated image analysis can greatly increase the granularity of this technology [77,78], with more intensity levels and (spatial) distributions without intra-/or interobserver variabilities. This information may then be used in combination with patient information and genetic data to identify risk categories or select patients for individual treatment options [79,80].

#### 2.2.2. Advanced Image Analysis Using Deep Learning

The emerging field of deep learning has revolutionized data science and image analysis in particular [81,82,83]. This has led to the emergence of radiomics, a field of radiology, where deep learning is used to detect genetic alterations by analyzing regular radiological data to predict patient and treatment outcome [84,85]. On the contrary, analyzing histopathological image data captures data of even higher granularity. Here, H&E is the standard staining used for tissue samples that is applied in daily routine pathological diagnostics. Detection of cancer traits, including relevant prognostic information, molecular subtypes and even origins of cancers have been proposed as proof-of-concepts using regular H&E brightfield images [86,87,88,89]. Images may be used to detect cellular features, like cell types and (anatomical) structures, that may be illustrated to a human reader or provided as prognostic biomarkers or as decision support algorithms [90,91]. What seems especially interesting is that the visual confirmation of results is of high interest to specialties like pathology, to mimic their working routine of interpretation of histological images [92]. In detail, regions-of-interest are highlighted, and therefore the algorithm is used to provide information where an observer can make an informed decision. However, there are also classifications that may assign classes to virtual-whole-slide imaging objects for histological subtypes and mutational status [86].

#### 2.2.3. Spatial Multiplex Analyses

Previous multiplexing fluorescent dyes allowed a given assay to assess a handful of antibodies in combination to phenotype subpopulations of cell types. Recently, with the advancement of technology, almost hundreds of antibodies may be combined. Technically, high-plex multicolor FACS machines may be used to identify cell populations from blood sample, including circulating tumor cells.

Several technologies have been developed to uncover spatial resolutions of cell types, which are a great resource to resolve immunological host responses to solid tumors [93]. Co-detection by indexing (CODEX), which uses DNA-conjugated antibodies, allows simultaneous overview of up to 60 markers in situ [94]. However, these data may need to be captured, stored, and analyzed in combination. On one hand, this requires analytical pipelines to identify cellular subpopulations, but also specific datatypes, embedded in medical data warehouses [95,96].

#### 2.2.4. The Important Role of Digitalization in the Medical Field

As the COVID-19 pandemic has revealed within multiple medical disciplines, digitalization is the key to allow a rapid exchange of relevant information. Given the emerging role of cancer diagnostics in pathology, where specific genetic alterations are revealed to guide treatment options at an individual level, it is puzzling that within this discipline there appears to be a lack of fundamental concepts in digitalization and standards of data formats [97]. Despite the obvious efforts in gathering infrastructural equipment allowing us to digitize a given histological slide to a virtual microscopy slide, allowing interpretation on a monitor, there seem to be more challenges ahead. Data and metadata need to be gathered, centrally, following standards in data formats, enabling them to be processed by high-performance computing and accessed by different disciplines.

What seems particular worrying is that most pathology departments run information systems that are not fully integrated into the hospital information system. Therefore, although information of high granularity may be accumulated, combining them with medical records requires more advanced data management systems. Although there are great opportunities for machine learning, results of machine learning models need to be transferred to pathology reports. However, structured reporting in pathology is still at an early stage and before translation into daily routine practice [98,99,100,101,102].

## 3. Translation of Approaches

### 3.1. Interpretation of Genomic Data

Next-generation sequencing (NGS) entails identifying the genetic sequence of an individual in millions of reads, which are subsequently put together into a whole sequence. Potentially millions of alterations can be identified that vary between the tumor and the “reference sequence”. As most genetic variants have little to unknown pathogenic impact, they must be carefully examined and screened to filter out the few that are clinically significant. Genomic variants are often categorized into a five-point scale from benign to pathogenic, with intervening scores of likely benign, uncertain, and likely pathogenic, to indicate the possibility of the mutation being associated with the disease [103]. In general, benign variants are not reported. If a matching normal sample was used as the reference sequence, germline variants can also be reported following the guidelines in addition to the somatic acquired variants [103]. For reliable test findings, the generation of sequence data, variant calling, and variant interpretation are all crucial procedures.

### 3.2. Considerations for Data Analysis

Several components need to be considered when analyzing genomics and image analysis data. Here, we focus on quality issues related to genomics and image analysis data, which can be broadly divided into two categories: sequencing and image analysis.

#### 3.2.1. Sequencing, Quality and Standardization

Types of sample and the sample quality of genomics data are aspects that must be accounted for during the analysis process. Samples collected for sequencing include blood, buccal swab, biopsy or resected tissue [25]. In general, the quality of DNA isolated from blood is expected to be high. Meanwhile, differing quality could be seen depending on the origin of tumor tissues, preservation methods (formalin-fixed, snap-frozen), size of tissue, number of samples collected from a patient (resection/biopsy), tumor cellularity as well as the presence of necrosis or lymphocyte infiltration [25]. In particular, working with a single core biopsy significantly increases the risk of misprofiling due to ITH [62], which refers to the existence of distinct tumor cell populations with differing molecular and phenotypical characteristics within the same tumor. Unlike traditional bulk-tissue sequencing, single cell sequencing sampling constitutes for a set of new, unique challenges to be tackled. Due to the limited quantity of material at disposal for each cell, observations are often fraught with ambiguity. In order to compensate for this, amplification is employed, eventually leading to the introduction of technical noise to the data [104]. Additionally, batch effect is another prominent feature that can arise from sample preparation due to its sequencing procedures [105].

As for the sample quality, several factors play a role, such as sample collection, handling, storage and isolation methods for DNA/RNA/protein. Isolation method becomes especially of importance for RNA, as it can result in different RNA quantities. Big data analyses integrating transcriptomic data from various sources revealed batch variations, and one contributing factor may have been the type of isolation [106].

As relevant intricacies will significantly impact the results and interpretation of genomics, it is important for researchers and clinicians to take into consideration of the important quality parameters, such as sequencing and processing steps, as well as setting appropriate thresholds and coverage between different systems [22]. For example, various systematic mistakes can be made in every sequencing platform with the use of differing chemistries for sequencing, making a propelling case for one to be aware of the different capacities and limitations of various platforms and sequencers. Illumina platforms generally show high quantitative power and accurate base calling [107], but may illustrate a systematic mistake of calling stretches of G’s when sequenced with a two-color chemistry system [108].

Another parameter that needs to be defined to ensure data quality is the average coverage. Coverage signifies the number of unique reads that are included in a given sequence and it can be used to measure the accuracy of the generated data [22]. The general rule of thumb is that the higher the average coverage of the analyzed data, it will be more precise and be able to detect certain nucleotide variants that exist in lower allele frequencies, which should be considered when analyzing different datasets.

#### 3.2.2. Image Analysis, Quality and Standardization

With the emerging role of machine learning in pathology and its standard H&E sections at center of deep learning activities, standardization needs to be prioritized, including data formats, color, quality and analytical approaches [97,109,110,111,112,113,114]. Once a given algorithm detects cell types, subcellular structures or quantifies biomarkers, results or even coordinates may be stored, for instance, using a given file format of convenience like JSON, HDF5 or XML (Figure 1). However, this could easily increase requirements for sophisticated data storage and databases. Potentially, results could be stored as metadata within an image object.

### 3.3. Infrastructural Requirements

As technology advances and more efforts are made to generate precise data, it is only natural to assume that the amount of raw data increases. One option is to downscale data, although this may result in a loss of discriminative information, especially in image analysis, due to the loss of tissue context when using small, high-resolution tiles [100]. Thus, availability of sufficient computational equipment and data storage is one of the major infrastructural requirements, despite other challenges reviewed elsewhere [119,120]. Institutions would have to invest significant resources for computer servers and facilities, such as sequencers, scanners, and cloud platforms, for effective integration and exchange of data. Moreover, as the metadata will encompass confidential information, appropriate security measurements with which the data is encrypted and stored will be of importance [26,119,121].

Once data has become available for access, management and accurate interpretation of a huge and complex amount of computational data would be another requirement. Highly specific skills and knowledge are required for both genomics and image analyses, which suggests for the development of specialty-specific educational opportunities to be initiated and promoted (Figure 1).

### 3.4. Challenges to Data Integration and Econimcal Aspects

In most cases, clinical data are obtained in multiple medical centers and there are no guidelines as to which patient phenotype should be recorded. As a result, oftentimes the collected information is inconsistent and inaccurate. Standardized terms and measures must be actively communicated between clinicians and researchers at an early stage, so that all retrieved information can be used with the highest relevance. Another suggestion is for educational efforts to be put into defining phenotypic terms and measurement, such as following the Human Phenotyping Ontology project guideline [122].

Multiple sequencing techniques/platforms and the vast amount of information obtained using genomics also pose added challenges to data integration [123]. Early interactions between researchers and clinicians would be recommended to first of all select the best diagnostic test so that all relevant information are covered. Secondly, prioritizing relevant variants for a specific disease would allow for reduction of enormous amount of data to a reasonable volume [123,124]. This does not mean that researchers should only analyze the defined gene sets, for example. Rather, these prioritized gene lists that are known to be highly relevant to the specific clinical indication would help in guidance for an in-depth review and determination of coverage of depth. Establishing such communication lines and procedures would help to ensure that the interpretation is reliable and efficient.

In the event that combined efforts would allow for a more dedicated and integrated analysis, how would this be incorporated in the current clinical setting for a given patient? One suggestion would be to carefully examine the patient data before designing an individual therapy concept. In detail, the concept of applying these strategies would need to be integrated into the workflow of the clinical decision making process. At the same time, one needs to be aware that clinical decision making takes personal interests and reservations into account.

In summary, in order for efficient exchange and integration of data, standardization of data for both diagnostic modalities and clinical outcome data must be made. This consistent and uniform effort would allow for multi-diagnostic approaches, such as genomic and/or image analysis information, to be directly linked and compared to clinical data.

## 4. Precision Oncology in Clinical Setting

The concept of precision oncology uses comprehensive biomarker testing to exploit molecular or immunological vulnerabilities in tumors for selecting the most specific and effective therapy. Furthermore, clinicians may use precision oncology for identification of high-risk populations, prevention of cancer, early diagnosis, possibility of specific cancer diagnoses, exploration of best treatment options and assessment of treatment efficacy [125].

This is best demonstrated in non-small cell lung cancer (NSCLC), where the identification of molecular alterations in driver genes, such as EGFR mutations, led to the approval and rapid development of the small molecule tyrosine kinase inhibitors (TKIs) [126]. The effective and improved responses compared to those from traditional treatments, such as chemotherapy, led to the incorporation of precision oncology modalities in daily practice. As opposed to non-selective chemotherapies, all patients with advanced NSCLC who are eligible for treatment nowadays require rapid and comprehensive screening of actionable biomarkers for first-line therapy selection [127]. For actionable biomarkers, such as ALK rearrangement, IHC and fluorescence in situ hybridization (FISH) are regarded as the gold standard in routine clinical diagnostic [128]. IHC expands to biomarkers of immuno-oncology drug response, such as PD-L1 expression or mismatch repair status, which indicate the eligibility for anti-PD-1/PD-L1 therapy in certain solid tumors [129].

However, as the number of novel predictive markers accumulates (currently to more than 30 genomic alterations and immune markers in NSCLC) and diagnostic demands increase, NGS-based approaches, such as DNA- and RNA-sequencing or targeted panels, have shown to provide a more efficient, cost- and tissue-saving tumor analysis, allowing assessment of a large number of genes in one assay compared to the traditional single-gene assessment procedures in routine molecular pathology [130]. These emerging biomarkers, such as ERBB2 (3%), BRAF (2%), PIK3CA (1%), MAP2K1 (1%), and NRAS (1%), are typically labelled as “niche alterations”, meaning they occur at a less frequent rate [131]. Nevertheless, that is not to say that they are less important than the already established biomarkers. For instance, a subset of ALK-rearranged NSCLC with TP53 mutations was associated with higher risk of resistance to ALK inhibitors than those without TP53 mutations [132]. Additionally, KEAP1/NRF2 mutations in stage I-III NSCLC also indicate enhanced risk of local recurrence after radiotherapy [133]. In parallel, composite biomarkers—a combination of several biomarkers—identified patients with ALK alterations that may benefit from immunotherapy [134]. From a regulatory perspective, although individual biomarkers are approved as companion diagnostics, combination of those as composite biomarkers would require individual approval [135].

Adding on to the complexity of precision oncology comes into play the integration of data generated using different diagnostic modalities (histopathological and genomics data) with non-omics data, such as images (CT, PET-CT, MRI, X-ray, whole-slide) and clinical data. This is considered to be one of the most challenging tasks to enable the futurist modules. In order to perform multi-platform data integration, it would require certain pre-processing steps including normalization, noise filtration and selection of highly relevant features [136]. One possible solution to this would be to implement artificial intelligence in order to guarantee better sensitivity, specificity and efficiency. For example, a model that was trained using deep neural networks could link gene expression with drug response and predict drug response and survival [137,138]. Furthermore, Yu et al., was able to determine the patient survival rate by combining NGS data with histopathology data in lung cancer cohort [139]. Although artificial intelligence is still at its infancy in the biomedical field, it has the ability to play a crucial role, not only in the analysis of complex and various data sets, but in data integration as well.

As exhibited by NSCLC, the significance of checking for potential genetic abnormalities paved the way to new diagnostic possibilities. Furthermore, novel emerging biomarkers, non-omics data and integration of multi-datasets have been explored for diagnostic purposes. Therefore, it is of importance to note the crucial role of integration between conventional and novel techniques for both medical decision-making and screening purposes.

## 5. Summary, Future Perspectives and Conclusions

Clearly, medical practice in the future will be more personally tailored. Our understanding of the genomic and molecular underpinnings of cancer growth, progression and resistance has been revolutionized fortunately due to the advancement of technology. Although genomic and image data are useful on their own, their integration is a crucial factor to the future of precision medicine. There are still several hurdles and issues that need to be addressed to fully receive the benefit of significant advances in genomic and molecular research in the pursuit of individualized approaches to clinical medicine. Appropriate measurements and initiation of actions will have to be supported to provide the basic ingredient of modern oncology.

## Figures and Tables

**Figure 1 biomolecules-11-01310-f001:**
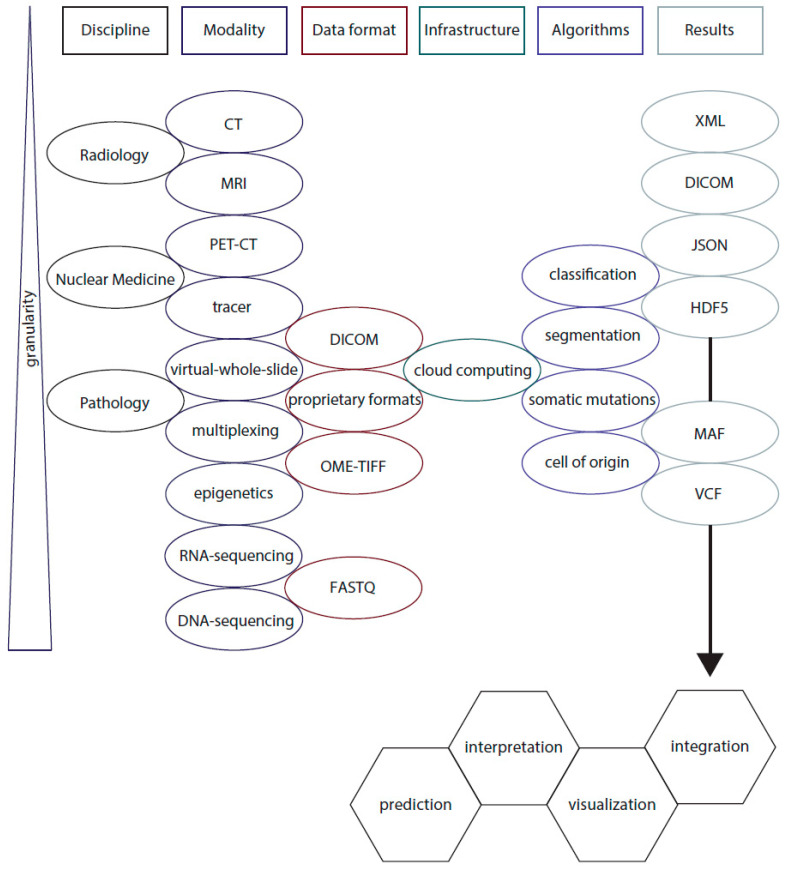
Schematic of integrative data analysis in oncology. Diagnostic medical disciplines, including Radiology, Nuclear Medicine and Pathology are gathering data using their different modalities, like computer tomography (CT), magnetic resonance imaging (MRI), positron emission tomography (PET-CT), as well as tracers that are able to highlight molecules of interest. While Radiology and Nuclear Medicine share a sophisticated data format of digital imaging and communications in medicine (DICOM) objects, which includes relevant (pseudonymized) patient (meta) data, other formats from virtual-whole-slide images and multiplexing techniques may be stored using proprietary vendor formats. The maturing file format of open microscopy environment tagged file format (OME-TIFF) appears to be especially interesting as future solution, as DICOM objects for virtual-microscopy-images have not yet been broadly accepted by the community [97,112,114]. Cloud computing services would require specialized structures, like central processing units (CPU) and graphical processing units (GPU) that meet the desired calculating capabilities. In the near future, several specialized algorithms will be executed allowing segmentation of regions of interest and classification of image objects, in addition to accurate variant calling and methylome-based cell of origin determination using epigenetics. The given results of image analysis are stored using extensible markup language (XML), DICOM objects or JavaScript Object Notation (JSON), together with Hierarchical Data Format (HDF5). Here, spatial descriptive statistics of cellular and subcellular structures of interest are calculated. Data formats like variant calling files (VCF) and Mutation Annotation Format (MAF; https://docs.gdc.cancer.gov/Data/File_Formats/MAF_Format/, accessed on 3 September 2021) may be used to store results of DNA-sequencing. Finally, the data need to be integrated and visualized [115,116] to allow interpretation in order to allow patient stratification, identifying risk groups of cancer patients and to predict therapy responses. Given the increasing complexity within oncology, analytical tools are needed to retrieve current information on drug targets, clinical trials and potential treatment options, which may need to be integrated in a given workflow [117,118].

**Table 1 biomolecules-11-01310-t001:** Summary and comparison of various diagnostic modalities.

Technique	Description	Platform	Data Analysis	Pros	Cons
Whole-genome sequencing (WGS)	whole genome is analyzed	Illumina PacBio Complete Ion Torrent BGI/MGI Oxford Nanopore	sequenced reads as data output with read alignments or quality scores variant identification annotation visualization statistical analysis	whole genomic sequence can be analyzed can identify non-coding mutations	costly and time-consuming for data interpretation high chance of incidental findings
Whole-exome sequencing (WES)	entire exome is analyzed			cost-effective and time-efficient than WGS deep coverage in exonic regions	high risk of incidental findings information only on coding regions
Targeted gene panel	captures key genes or regions of interest set by prior knowledge			significant reduction in time and cost compared to WGS/WES suitable as a diagnostic modality	requires prior knowledge of targeted regions not suitable for biomarker discovery
RNA-sequencing (RNAseq)	number of mRNA or total RNA molecules in the transcriptome is directly sequenced and quantified			can detect novel transcripts, fusions, single-nucleotide variants, indels, alternative splicing, allele-specific expression and newly transcribed regions good for biomarker discovery	need high-quality RNA (RNA integrity number > 8) only the expressed markers can be detected, thereby missing alterations in regulatory regions or non-expressed genes
Multiplex gene expression panel	a variation on RNA microarrays that uses hybridization probes	NanoString QuantiGene Plex	color-coded probes are converted into counts counts are normalized using housekeeping genes	RNA from FFPE material can be used can be done with less amount of RNA compared to RNAseq amplification free minimal background signal	not suitable for biomarker discovery limited flexibility
Epigenetic techniques	heritable phenotypical alterations that do not involve DNA sequence	Illumina Nimblegen Axon Roche	different epigenetic techniques are integrated based on these annotations, epigenome differences are recognized	epigenetic changes, such as DNA methylation or histone modification can be assessed	risk of variations depending on time of harvest and different organs/samples difficulty in choosing the techniques depending on the modification
Proteomic techniques	quantifies protein/peptide abundance, modification and interaction	mass spectrometry-based protein microarray-based	covalent changes are quantified by determining the equivalent change in protein mass in contrast to the unmodified peptide	gives a different level of understanding from D/RNA sequencing by high-throughput analyses of thousands of proteins in cells or body fluids	weak reproducibility and repeatability compared to other genomics techniques
Immunohistochemistry (IHC)	detection of molecules using antibodies, enzymatic/fluorescent dyes used to visualize by secondary antibody conjugates	automated staining devices from several suppliers dedicated multiplexing techniques e.g. Roche DISCOVERY / PerkinElmer Opal™	brightfield IHC requires microscopes while more advanced image analysis requires whole-slide-scanners fluorescent dyes require specialized imaging devices	allows spatial distributions of cell types / molecules of interest	limited to single / multiple molecules of interest on a given slide and therefore requires predefined antibody panels
*In situ* hybridization (ISH)	hybridization of RNA/DNA molecules using fluorescent (F-ISH) or chromogenic (C-ISH) dyes	RNAscope® Technology for RNA DNAscope™ for DNA	brightfield microscopes for CISH and fluorescent imagers for FISH application of object detection and sementic segmentations allows automation and quantitative analysis	quantitative measure of RNA/DNA molecules at cellular level can be applied on Formalin-Fixed Paraffin-Embedded (FFPE) tissues	limited to single / multiple molecules of interest
Single cell sequencing (sc-seq)	measures DNA, RNA, epigenetic marks and protein at a single-cell resolution	Illumina Ion Torrent BGI/MGI 10X Genomics	annotations of individual cells using cellular barcodes rest are analyzed in a similar manner to bulk sequencing	provides more precise classification of cell types and states than bulk sequencing	introduction of noise due to experimental procedures computational burden due to high dimensionality data difficult to integrate data from various types of single-cell approaches

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
