# Peer review of "Shifting Gears in Precision Oncology—Challenges and Opportunities of Integrative Data Analysis"

_biomolecules, 2021, doi:10.3390/biom11091310_

Round 1

Reviewer 1 Report

General comments: 

It is a well written article that gives good overview of the diagnostic modalities in relation to precision oncology. However, there are few additional things that need to be discussed and correct before the review is ready for publication. In oncology the genomic and proteomic biomarkers play an important role in drug development and subsequently to guide therapy decision in the clinic, thus I will suggest that these aspects briefly are discussed in the review.   

Specific comments:

Introduction, line 36-37

You write: “As a result, inhibition of Her2/neu in cases with amplification of ERBB2 resulted in clinical responses [4].”

Comment: In the referred study [4] by DJ Slamon et al. (N Engl J Med 2001; 344: 783-92) the patients were not selected based on amplification of the HER2 gene but based on overexpression of HER2 protein by IHC (IHC2+ or IHC3+). This must be corrected. Please also see Front. Oncol. 2021; 11: 676939. 

DNA-sequencing, line 91-92.

You write. “One of the most widely used area is pharmacogenomics, which analyzes how an individual’s genome will affect the therapeutic response [24].” 

Comment: I agree with your that a large number of genomic and proteomic biomarkers acts as predictors for a number of targeted anti-cancer drugs and in this context I will strongly recommends that you briefly describe the concept of companion diagnostics, which now is defined by both the FDA and the EMA (https://www.fda.gov/medical-devices/in-vitro-diagnostics/list-cleared-or-approved-companion-diagnostic-devices-in-vitro-and-imaging-tools).

Table 1. Summary of genomic technics

Please update Table 1. I will suggest that you split the table into two, one that covers the genomic and one for proteomic techniques. In relation to genomics, you must include the different ISH technologies and for proteomics you must also include IHC, especially as you later on in the review included image analysis of IHC slides. Furthermore, these technologies frequently are used in relation to predictive biomarkers/companion diagnostics for a number of targeted drugs (Transl Oncol. 2021: 14:101063).

Proteomics, line 156-158.

You write: “Although broad proteomic analysis is still an emerging field in precision oncology, many of these signaling pathways are already established as U.S. Food and Drug Administration (FDA)-approved biomarkers [19,50,51].”

Comments: Please include reference to the FDA List of Cleared or Approved Companion Diagnostic Devices (In Vitro and Imaging Tools) (https://www.fda.gov/medical-devices/in-vitro-diagnostics/list-cleared-or-approved-companion-diagnostic-devices-in-vitro-and-imaging-tools). In relation to proteomics, you must also briefly discuss IHC and I will also strongly recommend that you here or in another place discuss some of the point of views presented in Cell 2021; 184: 1662-70. 

The important role of digitalization in the medical field, 238-256.

Comments: I will strongly suggest that you also here discuss the concept of composite biomarkers.  

Text to Figure 1, 327-351.

Comments. There seems to be something wrong with layout. Please correct.

Author Response

(Authors replies in bold, reviewer comments in italic)

We would like to thank the reviewer for his/her comments on our manuscript. We think the comments are useful, detailed and structured and allowed us to improve the review significantly. Thank you again for this effort.

Introduction, line 36-37

You write: “As a result, inhibition of Her2/neu in cases with amplification of ERBB2 resulted in clinical responses [4].”

Comment: In the referred study [4] by DJ Slamon et al. (N Engl J Med 2001; 344: 783-92) the patients were not selected based on amplification of the HER2 gene but based on overexpression of HER2 protein by IHC (IHC2+ or IHC3+). This must be corrected. Please also see Front. Oncol. 2021; 11: 676939. 

We thank the reviewer for his/her comment on this matter. We have corrected the mistake and changed the sentence to overexpression.

DNA-sequencing, line 91-92.

You write. “One of the most widely used area is pharmacogenomics, which analyzes how an individual’s genome will affect the therapeutic response [24].” 

Comment: I agree with your that a large number of genomic and proteomic biomarkers acts as predictors for a number of targeted anti-cancer drugs and in this context I will strongly recommends that you briefly describe the concept of companion diagnostics, which now is defined by both the FDA and the EMA (https://www.fda.gov/medical-devices/in-vitro-diagnostics/list-cleared-or-approved-companion-diagnostic-devices-in-vitro-and-imaging-tools).

Thank you for the insight. We have added references to companion diagnostics and mentioned them within the introduction (lines 40-42).

Table 1. Summary of genomic technics

Please update Table 1. I will suggest that you split the table into two, one that covers the genomic and one for proteomic techniques. In relation to genomics, you must include the different ISH technologies and for proteomics you must also include IHC, especially as you later on in the review included image analysis of IHC slides. Furthermore, these technologies frequently are used in relation to predictive biomarkers/companion diagnostics for a number of targeted drugs (Transl Oncol. 2021: 14:101063).

Thank you for the suggestion. As proposed by the reviewer, we have added both IHC and ISH into the table. We have discussed to split the table into two, however, considering the fact that the reader will be able to browse through the different techniques more comprehensively altogether, we have decided to leave it as one.

Proteomics, line 156-158.

You write: “Although broad proteomic analysis is still an emerging field in precision oncology, many of these signaling pathways are already established as U.S. Food and Drug Administration (FDA)-approved biomarkers [19,50,51].”

Comments: Please include reference to the FDA List of Cleared or Approved Companion Diagnostic Devices (In Vitro and Imaging Tools) (https://www.fda.gov/medical-devices/in-vitro-diagnostics/list-cleared-or-approved-companion-diagnostic-devices-in-vitro-and-imaging-tools). In relation to proteomics, you must also briefly discuss IHC and I will also strongly recommend that you here or in another place discuss some of the point of views presented in Cell 2021; 184: 1662-70. 

We strongly agree and have included the review as a reference for potential reference for a given reader (lines 234-246) and added a reference to companion diagnostics (lines 40-42).

The important role of digitalization in the medical field, 238-256.

Comments: I will strongly suggest that you also here discuss the concept of composite biomarkers.  

We agree with the reviewer and have added the definition of companion diagnostics and an example in the manuscript (lines 602-615).

Text to Figure 1, 327-351.

Comments. There seems to be something wrong with layout. Please correct.

Thank you for the keen observation. We have corrected the legend of Figure 1 according to the style of the journal.

Reviewer 2 Report

The article reviews different technologies that must be integrated in the future to achieve the goal of better precision medicine. The article is well structured and interesting, because the review try to reinforce the need of integration of multiple platform to advance in Precision Oncology, and is well  written and structured (strengths), but the review of each individual topic is obviously shallow (weakness).  For this reason, more than make specific suggestions for each topic, I suggested that the authors include an additional chapter focus on an specific cancer type in which they can discuss the application and integration of different technologies.

Author Response

(Authors replies in bold, reviewer comments in italic)

We thank the reviewer for his/her comment. In order to increase the depth of the manuscript, we have added an additional chapter revolving around lung cancer (4. Precision oncology in clinical setting). We hope that this will provide more insight to the readers and make a compelling case for the importance and need for integration of different diagnostic modalities into current clinical settings. 

Reviewer 3 Report

The authors reviewed about omics data analysis and integrate data analysis for precision medicine. The manuscript was sophisticated and it would be insightful for readers.

Author Response

We thank the reviewer for reviewing our work.

Round 2

Reviewer 1 Report

I have no further comments